# Classification of *Salmonella enterica* of the (Para-)Typhoid Fever Group by Fourier-Transform Infrared (FTIR) Spectroscopy

**DOI:** 10.3390/microorganisms9040853

**Published:** 2021-04-15

**Authors:** Miriam Cordovana, Norman Mauder, Markus Kostrzewa, Andreas Wille, Sandra Rojak, Ralf Matthias Hagen, Simone Ambretti, Stefano Pongolini, Laura Soliani, Ulrik S. Justesen, Hanne M. Holt, Olivier Join-Lambert, Simon Le Hello, Michel Auzou, Alida C. Veloo, Jürgen May, Hagen Frickmann, Denise Dekker

**Affiliations:** 1Bruker Daltonik GmbH, 28359 Bremen, Germany; miriam.cordovana@bruker.com (M.C.); norman.mauder@bruker.com (N.M.); markus.kostrzewa@bruker.com (M.K.); 2Institute for Hygiene and Environment, 20539 Hamburg, Germany; andreas.wille@hu.hamburg.de; 3Department of Microbiology and Hospital Hygiene, Bundeswehr Central Hospital Koblenz, 56070 Koblenz, Germany; sandrarojak@bundeswehr.org (S.R.); ralfmatthiashagen@bundeswehr.org (R.M.H.); 4Operative Unit of Microbiology, IRCCS-Azienda Ospedaliero Policlinico Sant’Orsola-Universitaria di Bologna, 40138 Bologna, Italy; simone.ambretti@aosp.bo.it; 5Risk Analysis and Genomic Epidemiology Unit, Istituto Zooprofilattico Sperimentale Della Lombardia e dell’Emilia-Romagna, 43126 Parma, Italy; stefano.pongolini@izsler.it (S.P.); laura.soliani@izsler.it (L.S.); 6Department of Clinical Microbiology, Odense University Hospital, 5000 Odense C, Denmark; Ulrik.Stenz.Justesen@rsyd.dk (U.S.J.); Hanne.Holt@rsyd.dk (H.M.H.); 7Department of Microbiology, Université de Caen, Normandie, CEDEX 5, 14032 Caen, France; olivier.join-lambert@unicaen.fr (O.J.-L.); Lehello-s@chu-caen.fr (S.L.H.); auzou-m@chu-caen.fr (M.A.); 8University Medical Center Groningen, Department of Medical Microbiology and Infection Prevention, University of Groningen, 9700 AB Groningen, The Netherlands; a.c.m.veloo@umcg.nl; 9Infectious Disease Department, Bernhard Nocht Institute for Tropical Medicine Hamburg, 20359 Hamburg, Germany; may@bnitm.de or; 10University Medical Center Hamburg-Eppendorf (UKE), Tropical Medicine II Hamburg, 20359 Hamburg, Germany; 11Department of Microbiology and Hospital Hygiene, Bundeswehr Hospital Hamburg, 20359 Hamburg, Germany; frickmann@bnitm.de or; 12Institute for Medical Microbiology, Virology and Hygiene, University Medicine Rostock, 18057 Rostock, Germany; 13German Centre for Infection Research (DZIF), Hamburg-Lübeck-Borstel-Riems, 38124 Braunschweig, Germany

**Keywords:** FTIR-spectroscopy, *Salmonella*, Typhi, Paratyphi, *Salmonella* typing, IR Biotyper, diagnostics, test evaluation

## Abstract

Typhoidal and para-typhoidal *Salmonella* are major causes of bacteraemia in resource-limited countries. Diagnostic alternatives to laborious and resource-demanding serotyping are essential. Fourier transform infrared spectroscopy (FTIRS) is a rapidly developing and simple bacterial typing technology. In this study, we assessed the discriminatory power of the FTIRS-based IR Biotyper (Bruker Daltonik GmbH, Bremen, Germany), for the rapid and reliable identification of biochemically confirmed typhoid and paratyphoid fever-associated *Salmonella* isolates. In total, 359 isolates, comprising 30 *S*. Typhi, 23 *S*. Paratyphi A, 23 *S*. Paratyphi B, and 7 *S*. Paratyphi C, respectively and other phylogenetically closely related *Salmonella* serovars belonging to the serogroups O:2, O:4, O:7 and O:9 were tested. The strains were derived from clinical, environmental and food samples collected at different European sites. Applying artificial neural networks, specific automated classifiers were built to discriminate typhoidal serovars from non-typhoidal serovars within each of the four serogroups. The accuracy of the classifiers was 99.9%, 87.0%, 99.5% and 99.0% for *Salmonella* Typhi, *Salmonella* Paratyphi A, B and *Salmonella* Paratyphi C, respectively. The IR Biotyper is a promising tool for fast and reliable detection of typhoidal *Salmonella*. Hence, IR biotyping may serve as a suitable alternative to conventional approaches for surveillance and diagnostic purposes.

## 1. Introduction

*Salmonella enterica*, both typhoidal and nontyphoidal *Salmonella* are amongst the most important bacteria isolated from patients with bacteraemia in resource-limited countries [1,2,3,4,5]. While severe invasive disease due to enteric *Salmonella* has frequently been associated with young age and immunocompromised states [4,6], typhoid fever-associated *Salmonella* are also affecting immunocompetent individuals [7,8] and seem to be more prevalent in adult patients [7,8]. In febrile patients in tropical endemicity settings, typhoid fever is an important differential diagnosis to malaria [9], although co-infections are predominantly observed for non-typhoidal *Salmonella* [10].

The Paratyphi A-C serovars are typically associated with clinically less severe disease as opposed to infections caused by the serovar Typhi [11]. While *Salmonella* Typhi (*S.* Typhi) has a worldwide distribution in tropical areas and is associated with poor hygiene conditions [9], areas of endemicity for paratyphoid fever are mainly Asia and Africa, with a particularly high incidence of 150 cases/100,000 persons years in China [12,13,14,15,16]. Until now, global screening and surveillance efforts still leave room for improvement [17]. *S.* Paratyphi B, for example, was previously shown to be clonally diverse [18,19,20]. Also, unlike *S.* Typhi for which the only reservoir are humans, *S.* Paratyphi A, and *S.* Paratyphi C have been found in domestic animals [21,22].

Due to the disease severity, both rapid and reliable differentiation are desirable [11]. Serotyping is regarded as the gold standard for differentiation of *Salmonella enterica* on serovar level [23], but the approach is laborious, requires multiple and expensive sera and experienced personnel. Therefore, attempts towards more easy-to-apply and rapid diagnostic tools have been launched. These include PCR [24,25,26,27,28,29,30,31,32] and loop-mediated isothermal amplification methods [33,34,35,36,37,38,39], both exhibiting sensitivity and specificity issues [38,39]. In addition, cross-reactions with serovars with high sequence homology like *S.* Paratyphi A and *S.* Weltevreden, *S.* Paratyphi B and *S.* Abony, as well as *S.* Paratyphi C and *S.* Choleraesuis [30,40] have been observed. Next to molecular approaches, MALDI-TOF-MS (matrix-assisted laser-desorption-ionization time-of-flight) has been applied to identify *Salmonella* Typhi [41]. However, MALDI-TOF MS technology is not a suitable tool for the identification of *Salmonella* on serovar level, as those approaches [41] were based on similarities on a clonal level.

Fourier Transform Infrared Spectroscopy (FTIRS) is commonly used in chemistry to determine the molecular composition of a wide range of sample types. As different biomolecules absorb IR radiation in a characteristic range of wavelengths (lipids 3000–2800 cm^−1^, proteins 1700–1500 cm^−1^, phospholipids/nucleic acids 1500–1185 cm^−1^, and polysaccharides 1185–900 cm^−1^) [42]. From each bacterial cell, a unique absorption spectrum will be produced by FTIR. This represents the bacteria’s specific fingerprint signature, reflecting its biomolecular content in correlation with its genetic information [43,44,45]. FTIRS has been successfully applied in many studies for the discrimination of bacteria at different taxonomic levels (genera, species, serogroup/type, and even at strain level). So far, it has been proven to be a simple, quick, high-throughput and cost-effective bacterial typing technique [46,47,48,49,50,51,52].

Due to high genetic diversity, *Salmonella enterica* represents a promising species to be investigated by FTIRS. Several research groups investigated FTIRS to discriminate *S. enterica* serotypes using multivariate analysis and different bacterial collections [53,54,55,56,57,58], but without specific regard to (para-)typhoid *Salmonella* so far.

In this study, we evaluated the application of the IR Biotyper (IRBT), an FTIRS commercially available system for microbial typing, for the identification of typhoid and paratyphoid fever-associated *Salmonella*. We analyzed a strain collection comprising *S.* Typhi, *S.* Paratyphi A-C, as well as other, partly phylogenetically closely related *Salmonella* serovars belonging to the serogroups O:2, O:4, O:7 and O:9. Cluster analysis was performed to investigate the potential discrimination between typhoidal and non-typhoidal serovars within the different O-serogroups. Artificial intelligence-based algorithms for the automated classification were developed. The classifiers aimed to discriminate the samples in “Paratyphi (A/B/C)/Typhi”/“non-Paratyphi (A/B/C)/Typhi”. These were built with spectra from cultures grown on Columbia Blood agar and were tested with spectra from cultures on different media.

## 2. Materials and Methods

### 2.1. Strain Collection

A total of 359 well characterized isolates (corresponding to 50 serovars) were included in the study. Of these, 26 belonged to the O:2 group, 222 to the O:4 group, 45 to the O:7 group and 66 to the O:9 group (Table 1).

The *S.* Paratyphi A-C strains were provided by the Bundeswehr Hospital Hamburg, Germany, the Bundeswehr Central Hospital Koblenz, Germany; the Institute for Hygiene and Environment, City of Hamburg, Germany, *S.* Typhi by the Regional reference Center for Enteropathogens “Istituto Zooprofilattico sperimentale della Lombardia e dell’Emilia-Romagna” (IZSLER), Parma, Italy, the remaining isolates by different centers (University Hospital Policlinico Sant’Orsola-Malpighi, Bologna, Italy; Caen Normandie University Hospital, Caen, France; University Medical Center Groningen, The Netherlands).

All isolates were identified at genus level by MALDI-TOF MS or biochemical methods (API 20E, bioMérieux, Marcy l’Etoile, France), followed by characterization at serotype level by classical methods (serotyping [59], Whole Genome Sequencing [60,61,62], PFGE [63], and PCR for S. Typhimurium [64,65,66]).

### 2.2. Sample Preparation

Bacterial strains were stored at −80 °C using the microbank system (Microbanks, PRO-LAB DIAGNOSTICS, Richmond Hill, Canada) before they were retrieved on Columbia sheep blood agar (CBA—Becton, Dickinson and Company, Sparks, MD, USA). Infrared spectra were acquired from strains sub-cultured on a selection of different media (CBA, Tryptose Soy Agar (TSA), Chocolate agar (CHO) and Mueller-Hinton agar (MHA), Becton, Dickinson and Company, Sparks, MD, USA) for 22 ± 2 h at 35 ± 2 °C.

Sample preparation was performed following the manufacturer’s instructions. Briefly, a 1 µl overloaded loop of bacterial colonies taken from the confluent part of the culture was resuspended in 50 µL of a 70% ethanol solution in an IR Biotyper suspension vial. After vortexing, 50 µL of deionized water was added, and the solution was mixed by pipetting. Fifteen µL of the bacterial suspension were spotted in three technical replicates onto the 96-spots silicon IR Biotyper target and dried for 15–20 min at 35 ± 2 °C. All isolates belonging to typhoidal serovars and to serovars represented by less than five isolates were measured in three independent biological replicates, to achieve a more balanced number of spectra for each serovar.

The quality control Infrared Test Standards (IRTS 1 and IRTS 2) of the IR Biotyper kit were resuspended in 90 µL of deionized water, then 90 µL of absolute ethanol were added and mixed. 12 µL of suspension were spotted in duplicate onto the IR Biotyper target, and let dry as described for the samples.

### 2.3. Spectra Acquisition and Analysis

IRTS 1 and IRTS 2 were measured as quality controls prior to sample spectra acquisition in each run.

Spectra acquisition, visualization and processing were performed in transmission mode in the spectral range 4000–600 cm^−1^ (mid-IR). Spectra were smoothed using the Savitzky–Golay algorithm over nine data points, and the second derivative was calculated. Spectra were then cut to 1300–800 cm^−1^ (the carbohydrates region), and vector-normalized, to amplify differences between isolates, and to correct variations related to spectra acquisition [64].

### 2.4. Cluster Analysis

The discriminatory power of IR Biotyper regarding the discrimination of (para-) typhoidal *S. enterica* serovars was investigated for each O-group. Principal Component Analysis (PCA) and Linear Discriminant Analysis (LDA) were performed applying the IR Biotyper Client software version 3.0.

### 2.5. Development of Automated Classifiers

Automated classifiers were built using machine learning algorithms to allow the automated classification of unknown samples by applying a marker model calculated on a set of training spectra. The classifiers were automated to be automatically applied during the measurement of the isolates’ spectra with predefined characteristics—here belonging to the genus *Salmonella*.

Artificial neural networks (ANNs), included in the IR Biotyper software, were applied to build the classifiers. Four classifiers were built, one for each O-group that included a typhoidal *Salmonella* serovar (O:2, O:4, O;7 and O:9) each, using spectra measured from Columbia blood agar cultures. The ANNs were trained with spectra from a minimal number of isolates to explore the robustness of the method (Table 2). The isolates used for training were the more diverse ones among the strains available for each serovar (observed by HCA and PCA) to include the highest spectral variance. For each isolate of the typhoidal serovars, all three biological replicates were included in the training set. In contrast, for the non-typhoidal serovars, only one of the three biological replicates was used (arbitrary selection of the second replicate). This selection was applied in order to have a more balanced number of spectra of the two groups (typhoidal vs. nontyphoidal) to avoid introducing biases in the training phase.

The classifiers were tested using (i) the remaining two biological replicates of the isolates included in the training set; (ii) the remaining isolates of the same serogroup; (iii) all isolates of the same serogroup but taken from different media.

The classifiers provide a result with a scoring system and “traffic light” colour coding, which warns the user about potential outliers or novelties. A “green score” result means that the sample spectrum is located within the spectral space of the training set. A “yellow score” result means that the sample is located at the periphery of the spectral space of the training set. A “red score” value means that the sample spectrum is located far from the samples included in the training set. Results with red scores were counted as “no classification possible”, while results with green or yellow scores, which were incorrectly classified, were counted as “misclassification”. For each classifier/dataset combination, the accuracy (defined as the ratio between the number of spectra correctly classified and the total number of spectra) and the error rate (defined as the ratio between the number of spectra misclassified and the total number of spectra) were calculated, as well as sensitivity (defined as the rate of target isolates correctly classified as target strains), specificity (defined as rate of the non-target strains correctly classified as non-target strains) and positive and negative predictive values.

The accuracy and error rate were calculated, also testing the classifiers with spectra of the same dataset grown on TSA, chocolate agar and Mueller-Hinton agar, in order to evaluate the potential versatility of the automated classifiers.

### 2.6. Ethical Clearance

Ethical clearance was not required because only reference strains were used, while neither patients nor patient-related data were included in the analysis.

## 3. Results

### 3.1. Cluster Analysis

In the serogroups O:2, O:7 and O:9, all the isolates belonging to the typhoidal serovars (*S.* Paratyphi A, *S.* Paratyphi C and *S.* Typhi isolates, respectively) were well separated from the remaining serovars of the respective O-groups. However, in the O:4 group, several sub-clusters among the *S.* Paratyphi B isolates were observed, with one being located in the same spectral space as a sub-cluster of *S.* Typhimurium—monophasic variant.

The results of PCA and LDA for the different O-groups are shown in Figure 1, Figure 2, Figure 3, Figure 4, Figure 5, Figure 6, Figure 7 and Figure 8, respectively. The two approaches were chosen to optimally visualize the clustering.

### 3.2. Automated Classifiers

The automated classifiers for *S.* Paratyphi A, *S.* Paratyphi C and *S.* Typhi showed an accuracy of 99%, and an error rate of 0%. Sensitivity and specificity were 100%.

The automated classifier for *S*. Paratyphi B showed an accuracy of 87%, and an error rate of 10.8%. Sensitivity was 77.8% (*n* = 4 *S.* Paratyphi B isolates were misclassified) and specificity was 94.8% (*n* = 9 *S.* Typhimurium monophasic variant and *n* = 1 *S.* Saintpaul isolates were misclassified). Three isolates were not classified (*n* = 1 *S.* subsp. salamae O:4 and *n* = 2 *S.* Typhimurium monophasic variant).

Results of classifiers testing, including the use of other media, are shown in Table 3. Best identifications were achieved with isolates from CBA media. Reliability did not significantly decrease when using TSA and CHO media, but was slightly affected with cultures grown on MHA.

## 4. Discussion

The study was performed to assess the suitability of IR biotyping for the rapid and easy-to-perform identification of typhoid and paratyphoid *Salmonella*. The applied algorithms allowed a highly reliable classification of *S.* Typhi, *S.* Paratyphi A, and *S.* Paratyphi C. S. Paratyphi A, which is antigenically closely related other *Salmonella* serovars such as *S.* Nitra, *S.* Kiel, and *S.* Koessen, did not represent a challenge to the diagnostic algorithm. This is comparable to PCR- and LAMP-based diagnostic assays [30,38,39,40].

In this study, the classifiers for distinguishing (para-)typhoidal from non-(para-)typhoidal serovars were based upon a dataset of already characterized strains. For use with unknown samples, the IR Biotyper classification should be performed hierarchically, i.e., a classifier first identifies the typhoidal O-serogroups, and a secondary layer of classifiers subsequently discriminates between the typhoidal serovar and non-typhoidal serovars, since the spectral features involved in the differentiation of either O-serogroups or serotypes differ. A similar approach was already described in previous studies [54,55,56,57,58]. Individual classifiers can be applied manually, or they can be bundled and automatically applied by dedicated software. However, *S.* Paratyphi B remains a challenge due to its high clonal diversity and variable associations with either systemic or enteric disease [17,18,19,20,21,22]. The associated diversity was reflected by intense sub-clustering of the assessed *S.* Paratyphi B isolates within the spectral dimension within the O:4-serogroup. While the majority of the clones were located separate from other serovars, in the spectral space, two sub-clones showed partial overlapping with some of the *S.* Typhimurium monophasic variant isolates. This resulted in misclassification of isolates of other serovars (mainly restricted to *S.* Typhimurium monophasic variant), associated with enteric disease in patients [67]. The specificity issues of IR biotyping, in particular for *S*. Typhimurium, is limiting the applicability of the approach because of the importance and frequency of *S.* Typhimurium in causing human disease [67]. This would require the confirmation of IR biotyping-based results within the O:4-subgroup using other approaches. Further investigations are needed to understand the underlying mechanisms for this phenomenon.

In spite of this limitation, rapid and easy-to-apply IR biotyping is likely to gain increased importance in the identification of (para-)typhoid *Salmonella* for diagnostic and surveillance purposes. Compared to expensive and laborious approaches like next-generation sequencing-based typing, the IR biotyping approach allows typing within few minutes at reagents costs of few cents as far as a biotyping device is available. However, as (para-)typhoid fever is predominantly prevalent in resource-limited settings [9,12,13,14,15,16] where biotyping devices are so far not readily available, the broader application of the introduced approach needs to be awaited.

Columbia blood agar, as the most commonly used media in clinical microbiology worldwide, is suitable media for the introduced IR biotyping approach. Based on our method evaluation, the following steps for future diagnostic application in suspected *S.* Typhi or *S.* Paratyphi A–C infections should be followed: bacterial material will be subjected to the IR-biotyping approach as outlined in the method. Diagnoses of *S.* Typhi, *S.* Paratyphi A, and *S.* Paratyphi C with a green traffic-light score in the IR-Biotyper software can be considered as confirmed, while profiles with lower scores should make use of alternative assays. Within the O:4-serogroup comprising *S.* Paratyphi B, identification based on IR-biotyping can only be considered as presumptively identified but would require alternative confirmatory diagnostic approaches.

The study has a number of limitations. The majority of *Salmonella* strains in this assessment included serovars most frequently found in clinical patient samples. However, we cannot exclude possible misclassifications by the presence of other rare serovars than the ones tested. Secondly, while more than 20 *S*. Typhi, *S.* Paratyphi A and *S.* Paratyphi B were available for the assessments, only seven *S.* Paratypi C strains were analysed, limiting the interpretation of the data from the O:7-serogroup. Thirdly, lacking information on the origin of the reference strains makes the interpretation of the observed clustering of the *S.* Paratyphi B strains difficult.

## 5. Conclusions

IR biotyping of strains proved to be reliable for *S.* Typhi, *S.* Paratyphi A and *S.* Paratyphi C. Within the O:4-serogroup, including *S.* Paratyphi B, alternative approaches should be added to confirm or exclude *S.* Paratyphi B in case of clinical suspicion. Therefore, in spite of the limitations, the use of IR-Biotyper spectra for the identification of (para-) typhoid fever-associated *Salmonella* is a milestone in the rapid and easy-to-apply discrimination within *Salmonella* serovars and by far exceeds the discriminatory potential of other spectrum-based strategies like MALDI-TOF-MS [41].

Accordingly, future diagnostic implementation of this method can be expected after receiving regulatory approval.

## Figures and Tables

**Figure 1 microorganisms-09-00853-f001:**
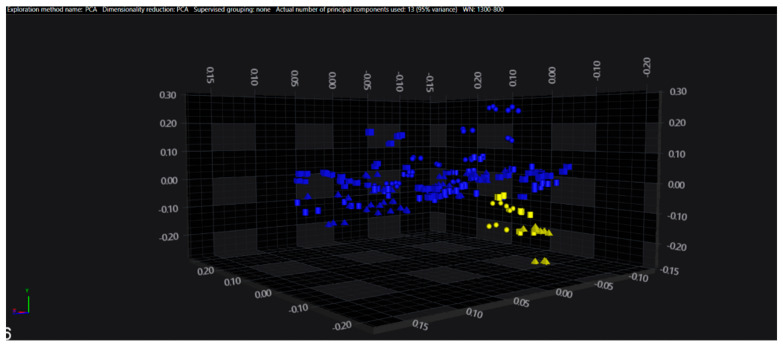
PCA of the O:2 serogroup (blue = Paratyphi A, yellow = other serovars)**.**

**Figure 2 microorganisms-09-00853-f002:**
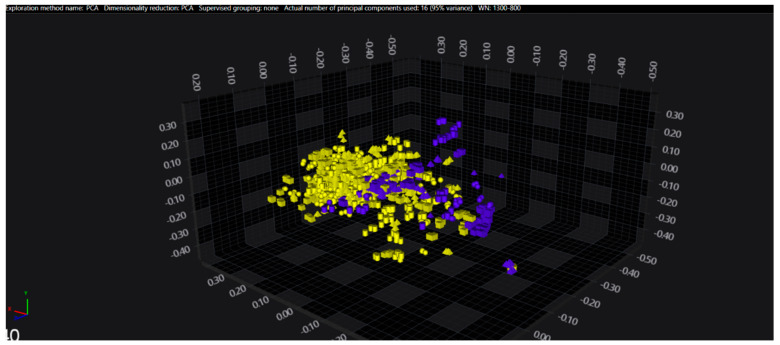
PCA of the O:4 serogroup (blue = Paratyphi B, yellow = other serovars).

**Figure 3 microorganisms-09-00853-f003:**
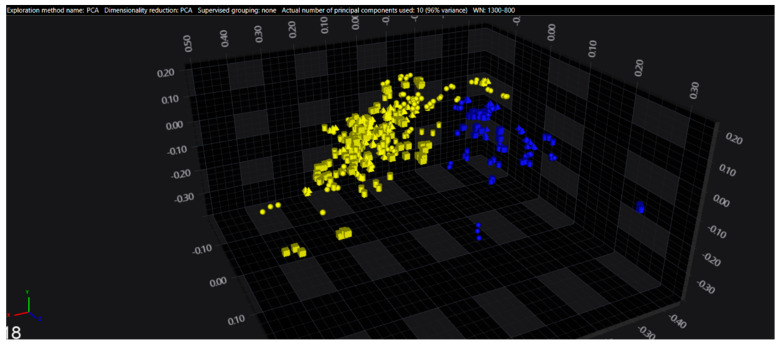
PCA of the O:7 serogroup (blue = Paratyphi C, yellow = other serovars).

**Figure 4 microorganisms-09-00853-f004:**
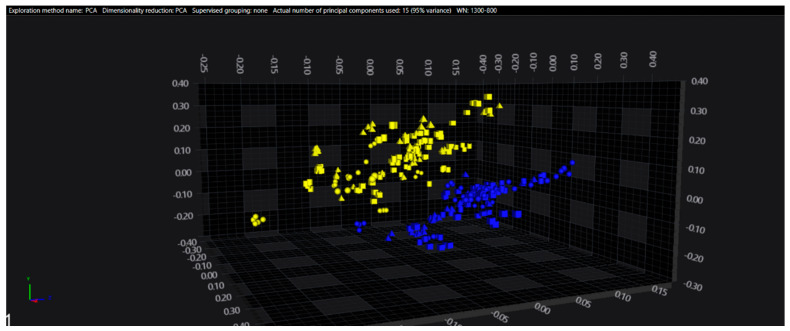
PCA of the O:8 serogroup (blue = Typhi, yellow = other serovars).

**Figure 5 microorganisms-09-00853-f005:**
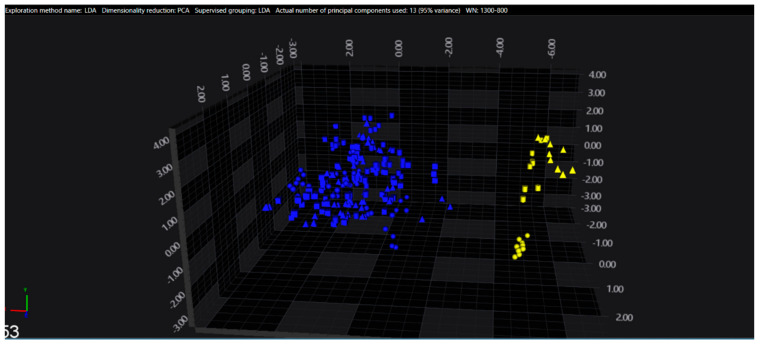
LDA of the O:2 serogroup (blue = Paratyphi A, yellow = other serovars).

**Figure 6 microorganisms-09-00853-f006:**
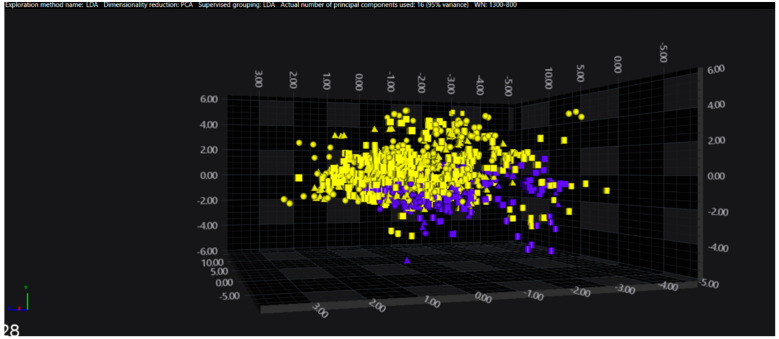
LDA of the O:4 serogroup (blue = Paratyphi B, yellow = other serovars).

**Figure 7 microorganisms-09-00853-f007:**
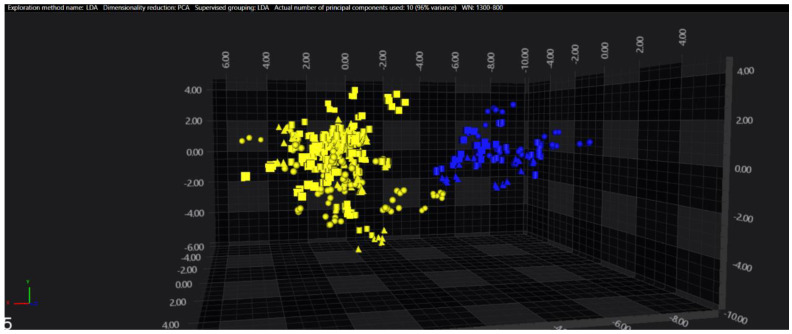
LDA of the O:7 serogroup (blue = Paratyphi C, yellow = other serovars).

**Figure 8 microorganisms-09-00853-f008:**
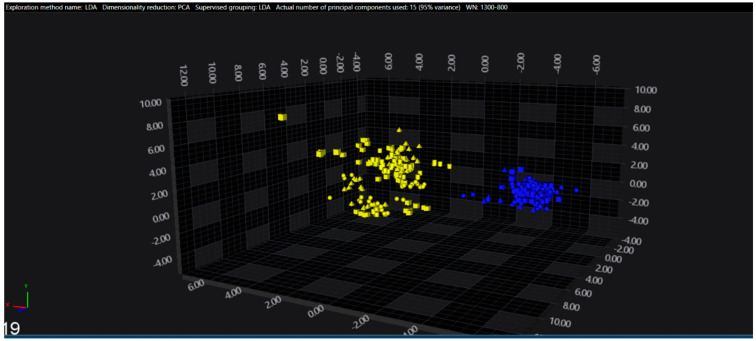
LDA of the O:9 serogroup (blue = Typhi, yellow = other serovars).

**Table 1 microorganisms-09-00853-t001:** *Salmonella enterica* isolates included in the analysis.

O-serogroup	Serovar	*N* = 359
O:2	Paratyphi A	23
	Kiel	1
	Koessen	1
	Nitra	1
O:4	Paratyphi B	23
	Aberdeen	1
	Abony	2
	Agona	6
	Banana	1
	Brandenburg	5
	Bredeney	1
	Chester	1
	Coeln	1
	Derby	8
	Heidelberg	2
	Hessarek	1
	Indiana	2
	Kisangani	1
	Paratyphi B var. Java	1
	Saintpaul	3
	Schleissheim	2
	Schwarzengrund	1
	Stanleyville	2
	Typhimurium	59
	Typhimurium (monophasic variant)	98
	Subsp. salamae O:4	1
O:7	Paratyphi C	7
	Braenderup	5
	Choleraesuis	2
	Infantis	11
	Isangi	2
	Livingstone	2
	Mikawasima	1
	Montevideo	3
	Ohio	2
	Oranienburg	3
	Singapore	1
	Strathcona	2
	Thompson	1
	Virchow	1
	Subsp. houtanae O:7	2
O:9	Typhi	30
	Dublin	1
	Enteritidis	19
	Javiana	1
	Kapemba	1
	Napoli	9
	Panama	2
	Zaiman	1
	Subsp. salamae O:9,12	2

**Table 2 microorganisms-09-00853-t002:** Characterization of the training and the testing sets.

O-groups	Training Set (*N*)	Testing Set CBA (*N*)	Testing Set TSA (*N*)	Testing Set CHO (*N*)	Testing Set MHA (*N*)
Paratyphi A (23)	5	18	23	23	23
Non-Paratyphi A (3)	3	3	3	3	3
Paratyphi B (23)	5	18	23	23	23
Non-Paratyphi B (199)	27	199	199	199	199
Paratyphi C (7)	2	5	7	7	7
Non-Paratyphi C (38)	14	37	38	38	38
Typhi (30)	5	25	30	30	-
Non-Typhi (36)	14	36	36	36	-

**Table 3 microorganisms-09-00853-t003:** Performance of the automated classifiers with different testing sets/media.

Agar Medium	Parameter	Classifier *S.* Paratyphi A	Classifier *S.* Paratyphi B	Classifier *S.* Paratyphi C	Classifier *S.* Typhi
CBA	Accuracy	99.5%	87.0%	99%	99.9%
	Error rate	0	10.8%	0	0
	Failed classification	0.5%	2.2%	1%	0.1%
	Sensitivity	100%	77.8%	100%	100%
	Specificity	100%	94.8%	100%	100%
	Positive predictive value ^#^	100%	77.8%	100%	100%
	Negative predictive value ^#^	100%	83.1%	100%	100%
TSA	Accuracy	97.0%	87.8%	95.5%	99.6%
	Error rate	1%	10.1%	2.8%	0
	Failed classification	2%	2.1%	1.7%	0.4%
CHO	Accuracy	100%	86.6%	98.8%	99.1%
	Error rate	0	12.0%	1.2%	0
	Failed classification	0	1.4%	0	0.9%
MHA	Accuracy	94%	84.9%	94.7%	-
	Error rate	0	11.7%	0	-
	Failed classification	6%	3.4%	5.3%	-

^#^ positive and negative predictive values are largely affected by prevalence, so these values may vary in other sample collections.

## Data Availability

All relevant data are provided in the manuscript. Raw data can be provided as reasonable request.

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
