# Peer review of "Classification of Salmonella enterica of the (Para-)Typhoid Fever Group by Fourier-Transform Infrared (FTIR) Spectroscopy"

_microorganisms, 2021, doi:10.3390/microorganisms9040853_

Round 1

Reviewer 1 Report

The authors completed and present a well-designed study that demonstrates the feasibility of FTIRS analysis of Salmonella serovars. The study is presented well, and the results are clear. I ask that the authors consider the following concerns and areas for improvement.

The statement of author responsibility leaves ambiguous the role of the majority of the listed authors. Aside from M.C., N.M., M.K., H.F., and D.D., it appears that the other authors only provided resources. The authors should review the authorship guidelines and ensure that all authors' contributions meet the guidelines for substantial contributions. If not, the contributors could be listed in the acknowledgements.

The statement on Data Availability does not address if the authors are making the training data set or the data related to the LDA classifier available. For other researchers to take advantage of this study for testing samples, these would be needed. If these are being integrated into a commercial product, that should be specified.

The authors do not address if a single classifier could distinguish the serogroups and serovars. This information would be valuable to those considering applying this as a testing technique.

The authors acknowledge that FTIRS has been applied to Salmonella previous in references [53-58], but no assessment is provided of whether the presented results are consistent with those studies or challenge them in some way.

The authors' reference to [41] creates some ambiguity about the effectiveness of MALDI-TOF. Publication [41] acknowledges that Salmonella MALDI-TOF databases do not provide this but presents research evidence that discrimination may be possible.

No clear explanation is offered of how the information in Figure 2 differs from that in Figure 1. An LDA analysis would typically only define a dividing surface and not further transform the data. The axis values (undefined) change and the angle of view appear to change, but nothing else.

Several formatting issues should be addressed by the authors or editors:

  • Line 128 incorrectly formats "°C"
  • Table 2 lacks a caption and, in the column headers, appears to use "n" and "N" interchangeably
  • Table 3 footnote repeats the definition of the Agar Medium abbreviations that was previously provided
  • Table 3 contains at least two "!" in place of "1"

Author Response

Thank you for your suggestions. Please find below our answers. Best wishes, Denise Dekker

The authors completed and present a well-designed study that demonstrates the feasibility of FTIRS analysis of Salmonella serovars. The study is presented well, and the results are clear. I ask that the authors consider the following concerns and areas for improvement.

  1. The statement of author responsibility leaves ambiguous the role of the majority of the listed authors. Aside from M.C., N.M., M.K., H.F., and D.D., it appears that the other authors only provided resources. The authors should review the authorship guidelines and ensure that all authors' contributions meet the guidelines for substantial contributions. If not, the contributors could be listed in the acknowledgements. Authors: We confirm that all listed authors provided substantial intellectual input justifying the listing as co-authors. In particular, all authors contributed by reviewing and editing the draft of the study at various stages, thus providing considerable scientific contributions.
  1. The statement on Data Availability does not address if the authors are making the training data set or the data related to the LDA classifier available. For other researchers to take advantage of this study for testing samples, these would be needed. If these are being integrated into a commercial product, that should be specified. Authors: We have added in the Data Availability Statement that raw data can be made available at reasonable request.
  1. The authors do not address if a single classifier could distinguish the serogroups and serovars. This information would be valuable to those considering applying this as a testing technique. Authors: Please refer to lines 281-289 for more information.

The authors acknowledge that FTIRS has been applied to Salmonella previous in references [53-58], but no assessment is provided of whether the presented results are consistent with those studies or challenge them in some way. Authors: we have now added that those studies were performed without specific regard to (para-)typhoid salmonellae. Please see line 97-98

  1. The authors' reference to [41] creates some ambiguity about the effectiveness of MALDI-TOF. Publication [41] acknowledges that Salmonella MALDI-TOF databases do not provide this but presents research evidence that discrimination may be possible. Authors: Indeed, there is no contradiction. The approaches described in [41] were just based on similarities on a clonal level as freely admitted by the authors of [41]. We have added this information to make the sentence easier understandable.
  1. No clear explanation is offered of how the information in Figure 2 differs from that in Figure 1. An LDA analysis would typically only define a dividing surface and not further transform the data. The axis values (undefined) change and the angle of view appear to change, but nothing else. Authors: The reviewer is right. As stated now (Results chapter, sub-heading 3.1. Cluster analysis, second paragraph, new last sentence), the two approaches were just chosen to optimally visualize the clustering.
  1. Several formatting issues should be addressed by the authors or editors: Line 128 incorrectly formats "°C". Authors: The typing error has been corrected.
  1. Table 2 lacks a caption and, in the column headers, appears to use "n" and "N" interchangeably. Authors: The heading “Characterization of the training and the testing sets” and the capital letter “N” has now been used consistently.
  1. Table 3 footnote repeats the definition of the Agar Medium abbreviations that was previously provided. Authors: As requested, the superfluous repetition of the agar medium abbreviations has been removed.
  1. Table 3 contains at least two "!" in place of "1" Authors: The typing errors have been removed as requested.

Reviewer 2 Report

In their study "Classification of Salmonella enterica of the (para-)typhoid fever group by Fourier-Transform Infrared (FTIR) spectroscopy" Cordovana et al. described the usage of the FTIRS-based IR Biotyper (Bruker Daltonik) for the identification of typhoid and paratyphoid fever-associated Salmonella isolates.

Overall, this study is very well written and of great interest for all people working on this field of study.

I have one question/one remark regarding the discussion:

What are the advantages of this method regarding costs, time for the analysis etc. in comparsion to other methods such as for example genome sequencing?

Author Response

Reviewer 2: Thank you for your comment. Please find below our answer. Best wishes, Denise Dekker

In their study "Classification of Salmonella enterica of the (para-)typhoid fever group by Fourier-Transform Infrared (FTIR) spectroscopy" Cordovana et al. described the usage of the FTIRS-based IR Biotyper (Bruker Daltonik) for the identification of typhoid and paratyphoid fever-associated Salmonella isolates.

Overall, this study is very well written and of great interest for all people working on this field of study.

I have one question/one remark regarding the discussion:

What are the advantages of this method regarding costs, time for the analysis etc. in comparsion to other methods such as for example genome sequencing?

Authors: As requested, we have now stated that compared to expensive and laborious approaches like next generation sequencing-based typing, the IR biotyping approach allows typing within few minutes at reagents costs of few cents as far as a biotyping device is available (Discussion, third paragraph, new second sentence). In the next sentence, however, we admit that the availability of biotyping devices in resource-poor setting is indeed still a challenge.